# BAG OF TRICKS FOR UNSUPERVISED TTS

**Yi Ren**[1], **Chen Zhang**[2,1], **Shuicheng Yan**[1]

[1]SEA AI Lab, [2]Zhejiang University
`rayeren613@gmail.com, zc99@zju.edu.cn, yansc@sea.com`

## ABSTRACT

Unsupervised text-to-speech (TTS) aims to train TTS models for a specific language without any paired speech-text training data in that language. Existing methods either use speech and corresponding pseudo text generated by an unsupervised automatic speech recognition (ASR) model as training data, or employ the back-translation technique. Though effective, they suffer from low robustness to low-quality data and heavy dependence on the lexicon of a language that is sometimes unavailable, leading to difficulty in convergence, especially in low-resource language scenarios. In this work, we introduce a bag of tricks to enable effective unsupervised TTS. Specifically, 1) we carefully design a voice conversion model to normalize the variable and noisy information in the low-quality speech data while preserving the pronunciation information; 2) we employ the non-autoregressive TTS model to overcome the robustness issue; and 3) we explore several tricks applied in back-translation, including curriculum learning, length augmentation and auxiliary supervised loss to stabilize the back-translation and improve its effectiveness. Through experiments, it has been demonstrated that our method achieves better intelligibility and audio quality than all previous methods, and that these tricks are very essential to the performance gain.

## 1 INTRODUCTION

Text to speech (TTS), or speech synthesis, has been a hot research topic (Wang et al., 2017; Shen et al., 2018; Ming et al., 2016; Arik et al., 2017; Ping et al., 2018; Ren et al., 2019a; Li et al., 2018; Ren et al., 2021a; Liu et al., 2021; Ren et al., 2021b) and has broad industrial applications as well. However, previous TTS has been developed dominantly for majority languages like English, Mandarin or German, while seldom for minority languages and dialects (low-resource languages), as supervised TTS requires hours of single-speaker and high-quality data to retain a good performance, but collecting and labeling such data for low-resource languages are very expensive and need a substantial amount of manpower.

Recently, some works exploit unsupervised (Ni et al., 2022; Liu et al., 2022b) or semi-unsupervised learning (Tjandra et al., 2017; Ren et al., 2019b; Liu et al., 2020; Xu et al., 2020) to enable speech synthesis for low-resource languages, some of which are summarized in Table 1. Semi-supervised methods rely on **a small amount of high-quality paired data in the target language** to initialize the model parameters and employ back-translation to leverage the unpaired data. But high-quality paired data in minor languages are usually collected via recording in professional studios or transcribing by native speakers, and hence very costly and sometimes even unaffordable to attain. In contrast, unsupervised methods train an unsupervised automatic speech recognition model (ASR) (Baevski et al., 2021; Liu et al., 2022a) to generate pseudo labels for the unpaired speech data, and then use the pseudo labels and speech paired data to train the TTS model. However, their performance tends to be **bounded by the performance of the unsupervised ASR model**, which is extremely difficult and unstable to train on some low-resource languages, especially for those without lexicon or grapheme-to-phoneme (G2P) tools (Baevski et al., 2021; Liu et al., 2022a)[1]. Besides,

---

[1]Baevski et al. (2021) claimed their method "requires phonemization of the text for the language of interest", and Liu et al. (2022a) claimed "when switching to an entirely letter-based system without a lexicon, the unit error rate increases substantially".

Table 1: Comparison of some semi-supervised and unsupervised TTS methods. "G2P" denotes grapheme-to-phoneme tool; "Paired (tgt)" and "Paired (other)" mean using paired data in the target language and other languages; "BT" denotes back-translation; "NAR" denotes non-autoregressive architecture for TTS model. "Semi." denotes semi-supervised and "Unsup." denotes unsupervised.

| Methods | Type | G2P | Dataset settings | | | | BT | NAR |
| | | | Multispeaker | Paired (tgt) | Paired (other) | Noisy | | |
| --- | --- | --- | --- | --- | --- | --- | --- | --- |
| Ren et al. (2019b) | Semi. | ✓ | ✗ | few | ✗ | ✗ | ✓ | ✗ |
| Xu et al. (2020) | Semi. | ✓ | ✓ | few | ✓ | ✓ | ✓ | ✗ |
| Liu et al. (2022b) | Unsup. | ✓ | ✓ | ✗ | ✗ | ✗ | ✗ | ✗ |
| Ni et al. (2022) | Unsup. | ✓ | ✗ | ✗ | ✗ | ✗ | ✗ | ✗ |
| Ours | Unsup. | ✗ | ✓ | ✗ | ✓ | ✓ | ✓ | ✓ |

the unpaired speech samples used in existing unsupervised methods are **clean** and ready for general TTS model training, such as CSS10 (Park & Mulc, 2019), LibriTTS (Zen et al., 2019) and LJSpeech (Ito, 2017). However, in real low-resource language scenarios, there is no guarantee that enough clean data can be obtained.

In this work, we aim to train an unsupervised TTS model in a low-resource language (the target language) with unpaired data, rather than any paired speech and text data, in that language, and also paired data in other rich-resource languages for initialization. Such training data are easily accessible. For example, the unpaired speech and text in the target language can be crawled from video or news websites in the countries using that language; the paired data in rich-resource languages can be obtained from some ASR and TTS datasets. Besides, these crawled speech data are from different speakers. Under such a task setting, we need to address the following challenges in order to achieve our goal. 1) **Low-quality multi-speaker data**. The speech data to be used for unsupervised training in our problem are often multi-speaker and low-quality, with much variable and noisy information like timbre, background noise, *etc.*, hindering model convergence and meaningful speech-text alignment. This significantly increases the difficulty of the TTS model training. 2) **Back-translation stability**. Previous semi-supervised TTS methods (Xu et al., 2020; Ren et al., 2019b) leverage the unpaired data with back-translation, but only achieving limited performance and sometimes difficult to converge, especially in unsupervised settings. 3) **Robustness**. Previous semi-supervised/unsupervised TTS methods (Xu et al., 2020; Ren et al., 2019b; Ni et al., 2022; Liu et al., 2022b) use an auto-regressive architecture (Li et al., 2018; Shen et al., 2018), which suffers from word missing and repeating issues, especially when the supervision signal is very weak. 4) **Lack of lexicon**. For low-resource languages, it is usually difficult to obtain existing lexicons or G2P tools.

We propose several practical tricks to address these issues and enable unsupervised TTS without any paired data in the target language and bridge the performance gap between the unsupervised and supervised TTS. Specifically, 1) we normalize the variable and noisy information in the **low-quality training data**. We propose a cross-lingual voice conversion model with flow-based enhanced prior, which converts the timbre of all sentences in different languages to one same speaker's voice while preserving the pronunciation information. 2) We explore some tricks including curriculum learning, length augmentation and auxiliary supervised loss to improve the effectiveness of **back-translation**. 3) To strengthen model **robustness**, we employ the non-autoregressive (NAR) TTS model and use the alignment extracted from the ASR model[2] in the back-translation process to guide the NAR TTS model training. By applying such a bag of tricks, we can successfully train an effective TTS model **with noisy and multi-speaker data** and **without any lexicons**.

Through experiments, it has been verified that our method can achieve both high-quality and high-intelligibility TTS, in terms of MOS and of word error rate (WER) and character error rate (CER) evaluated by external ASR, respectively. We compare our method to existing unsupervised TTS baselines (Ren et al., 2019b; Xu et al., 2020; Ni et al., 2022; Liu et al., 2022b) and find it significantly outperforms them in both audio quality and intelligibility under the same experimental settings. We conduct some analyses on the proposed tricks, which demonstrate the importance and necessity of

---

[2]The ASR model is the byproduct of back-translation, which does not need any extra paired data.

these tricks to achieve state-of-the-art unsupervised TTS. The samples generated by our models can be found at https://unsupertts-tricks.github.io.

## 2 RELATED WORKS

### 2.1 SUPERVISED SPEECH SYNTHESIS

In the past few years, with the development of deep learning, neural network-based TTS has thrived (Wang et al., 2017; Tachibana et al., 2018; Li et al., 2019; Ren et al., 2019a; 2021a; Łańcucki, 2020), where the text-to-speech mapping is modeled by deep neural networks using encoder-decoder architectures. Early methods by Wang et al. (2017) and Ping et al. (2017) generate the mel-spectrogram autoregressively. However, they suffer from slow inference and low robustness issues, *e.g.* word skipping and repeating. To tackle these issues, later works explore non-autoregressive (NAR) speech generation. FastSpeech (Ren et al., 2019a) is the first non-autoregressive TTS architecture, which adopts the duration predictor and length regulator to bridge the length gap between the speech and the text sequence. After that, many methods are proposed, such as FastSpeech 2 (Ren et al., 2021a), Glow-TTS (Kim et al., 2020) and EATS (Donahue et al., 2021), achieving not only better audio quality but also fast inference and good robustness. Recently, some NAR models leveraging variational auto-encoder (VAE) to model the variation information in the latent space are developed, like VITS (Kim et al., 2021b) and PortaSpeech (Ren et al., 2021b), and they quickly become popular. In this work, we also employ non-autoregressive architecture and VAE structure to achieve robustness against low-quality data.

### 2.2 LOW-RESOURCE SPEECH SYNTHESIS

Supervised speech synthesis requires high-quality paired speech and text data for training, which are costly to attain, especially for low-resource languages. To broaden the application scope of TTS systems, several low-resource TTS models are developed, which only need a few or even not any high-quality paired data. Instead, they use unpaired text and audio data to train TTS models in a semi-supervised or unsupervised way, which are much straightforward and cheap to obtain.

**Semi-supervised TTS.** Ren et al. (2019b) adopt back-translation and pre-training to leverage unpaired data, generating pseudo text/speech samples with ASR/TTS models and training them with the augmented data iteratively. However, as a proof of concept, Ren et al. (2019b) only verify the feasibility of semi-supervised TTS in a single-speaker dataset. Later, LRSpeech (Xu et al., 2020) supports multi-speaker and noisy datasets and is closer to real application. However, *these semi-supervised methods still require a few pairs of high-quality speech and text data*, which are expensive for low-resource languages since they often need to be recorded in professional studios.

**Unsupervised speech synthesis.** Unsupervised speech synthesis does not use any paired training data from the target speaker and language, which has attracted growing attention recently. As the earliest unsupervised TTS works, Liu et al. (2022b) and Ni et al. (2022) both use an unsupervised ASR model to transcribe the TTS speech data to pseudo text and train with the augmented data to build an unsupervised TTS system. However, they heavily rely on the unsupervised ASR technique, whose training procedure is very unstable and heavily relies on lexicons. Therefore, these methods are *difficult to apply to other low-resource languages*. Besides, when switching to a multi-speaker setup, the gap between supervised and these unsupervised TTS methods becomes larger than single-speaker setup (Liu et al., 2022b). A recent ArXiv paper (Lian et al., 2022) trains a non-parallel voice conversion model using unpaired speech data as the acoustic model and a specific module to map the text sequence to the speech discrete representation sequence, but this module has to be trained with an external dataset with the same language as in the unpaired dataset. Thus this method is hardly applicable to real low-resource language scenarios due to the difficulty of collecting such a large paired dataset in this language.

## 3 PROPOSED BAG OF TRICKS

Suppose we have an unpaired speech dataset $S_{low}$ and a text dataset $T_{low}$ in the target low-resource language $L_{low}$, together with a paired speech-text dataset $S_{rich}$ and $T_{rich}$ in another rich-resource

language $L_{rich}$ as auxiliary supervised training data. We assume that $L_{low}$ and $L_{rich}$ share some common characters, such as Indonesian and French share some Latin alphabets. $S_{low}$ is a multi-speaker speech dataset whose audio quality is extremely low, as it is difficult to obtain enough single-speaker clean data for the low-resource language. As for the auxiliary training data, since there are many public speech audios available in rich-resource languages, we do not impose any restrictions on the quality of these speech data. Our method aims to train the TTS model in the language $L_{low}$ using the above datasets. Besides, we need another clean speech dataset $S_{ref}$ to provide the target timbre in our voice conversion model and it can be part of $S_{rich}$ or $S_{low}$. In this section, we first describe the overall training pipeline. Then we introduce model designs and some tricks used in each stage of the pipeline.

### 3.1 OVERALL TRAINING PIPELINE

As shown in Figure 1, the training pipeline of our method consists of 3 stages: voice conversion, supervised warm-up training and unsupervised back-translation training. We put the detailed pseudo-code algorithm of our training pipeline in Appendix A and describe each stage in the following paragraphs.

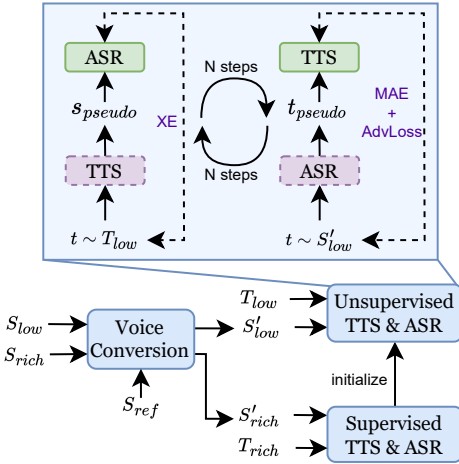

Figure 1: The overall pipeline of our method. The top part of the figure shows the iterative back-translation in our method. The modules marked in green with solid border are trainable and those in purple with dotted border are fixed. "XE" denotes cross-entropy loss; "MAE" and "AdvLoss" denote mean absolute error and adversarial loss.

**Stage 1: Voice conversion.** The low-resource speech dataset $S_{low}$ contains many speakers and can be very noisy. We consider the variable and noisy information, *e.g.*, background noise, speaker timbre, accent and some specific prosody, as the text-independent information in speech. Although some variable information is essential for certain TTS tasks like emotional, expressive and personalized TTS, it would be an obstacle for unsupervised TTS. The core purpose of unsupervised TTS is to solve the information matching problem between two modalities, *i.e.*, speech and text, which are actually aligned by the pronunciation (or called content). The variable information in speech may interfere with the crossmodal pronunciation information matching in the unsupervised training stage and make the model struggle to find aligned clues in the text for this variable information. Therefore intuitively, if we can reduce the information gap between speech and text, our TTS and ASR model can achieve crossmodal pronunciation information matching faster, and the unsupervised training process can then be stabilized. To this end, we apply the cross-lingual voice conversion as the first stage. We train the voice conversion model on the datasets $S_{low}$, $S_{rich}$ and a clean dataset $S_{ref}$ providing the reference speaker timbre and can be in any language. Then we can normalize the variable and noisy speech information of audios in $S_{low}$ and $S_{rich}$ using the voice conversion model and denote the generated datasets as $S'_{low}$ and $S'_{rich}$.

**Stage 2: Supervised warm-up training.** It is still very difficult to directly train unsupervised TTS from scratch even though we have normalized the variable information in the speech dataset $S_{low}$. To warm up the models for next unsupervised training, we train a sequence-to-sequence-based ASR model, which is required in the following back-translation stage, and a non-autoregressive TTS model using the auxiliary paired dataset $S_{rich}$ in a rich-resource language. This stage can provide a better initialization for the model since there exist certain commonalities between written and spoken formats in different languages[3].

**Stage 3: Unsupervised back-translation training.** Back-translation, originating from neural machine translation, is one of the most effective ways to leverage monolingual data for translation. In unsupervised TTS, back-translation (Sennrich et al., 2016; He et al., 2016; Ren et al., 2019b) leverages the dual nature (He et al., 2016; Qin, 2020) of TTS and ASR tasks and develops the capability of

---

[3]For a low-resource language $L_{low}$, it is usually not difficult to find a rich-resource language $L_{rich}$ which is close to $L_{low}$.

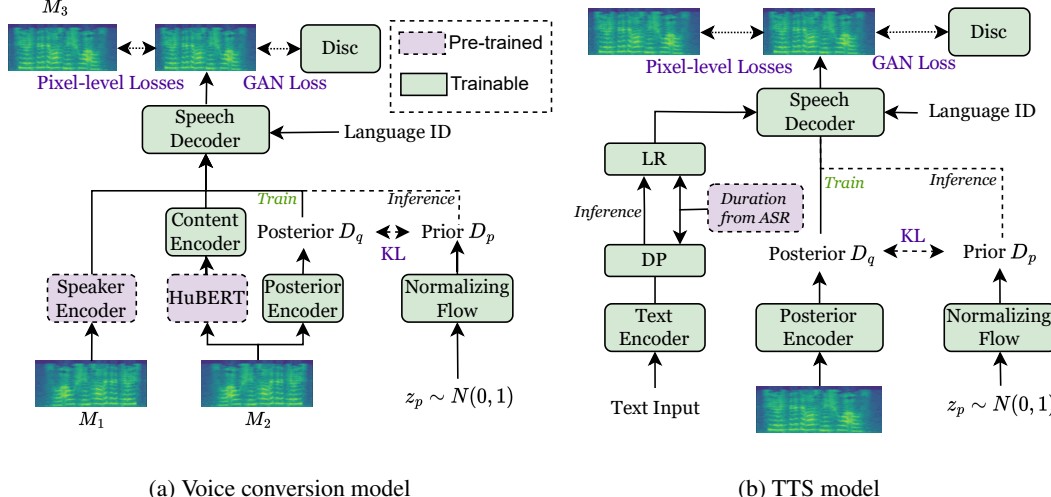

Figure 2: The voice conversion and TTS architecture in our method. In subfigure (a), "Disc" denotes discriminator; in training, $M_1$, $M_2$ and $M_3$ are all the same; in inference, $M_1$ is the content provider mel-spectrogram from the source speaker and $M_2$ is the target speaker's reference mel-spectrogram. In subfigure (b), "LR" and "DP" denote length regulator and duration predictor.

transforming text to speech (TTS) and speech to text (ASR). We transform a speech sequence $s$ into a text sequence $t_{pseudo}$ using the ASR model, and then train the TTS model on the transformed pair $(t_{pseudo}, s)$. Similarly, we also train the ASR model on the transformed pair $(s_{pseudo}, t)$ generated by the TTS model. The back-translation has two training directions, and as the training directions shift, the accuracy of ASR and performance of TTS can be boosted iteratively. We show the performance improvements as the back-translation training progresses in Appendix D.1.

## 3.2 VARIATIONAL VOICE CONVERSION MODEL WITH ENHANCED PRIOR

The new voice conversion (VC) model in Stage 1 is aimed at normalizing the variance and noisy information in low-quality audios. It is based on self-supervised learning (SSL) audio representation (Polyak et al., 2021; van Niekerk et al., 2022), which has been proved to be very effective in disentangling the content and timbre information. As shown in Figure 2a, the overall architecture of our VC model is like an autoencoder. In training, the mel-spectrogram $M_1$ and $M_2$, which are the same here, are fed into several information extraction modules to generate disentangled representations. Specifically, 1) the pre-trained *speaker encoder* extracts the sentence-level speaker embedding; 2) the pre-trained HuBERT (Hsu et al., 2021) extracts the frame-level SSL discrete representations containing content (pronunciation) information; and 3) the *posterior encoder* extracts the residual information. After the information decomposition, the speech decoder takes all the representations as input and reconstructs the mel-spectrogram using a mean absolute error (MAE) and a multi-length adversarial loss ($L_{adv}$) following Ye et al. (2022) and Chen et al. (2020). In inference, we replace $M_1$ with the reference speech which provides the target speaker timbre. In this way, the generated speech can preserve the pronunciation information in $M_2$ and transfer its timbre to $M_1$.

However, besides the common merits of general VC including preserving content and converting timbre, our model should also have the below properties to ensure its performance in our pipeline. 1) Our model should be **cross-lingual**. It should be able to normalize $S_{low}$ and $S_{rich}$ which are in different languages. 2) Our model should generate **high-quality** results. As the normalized speech will be fed to the next two stages as the training target, its result would bound that of the whole pipeline. 3) Our model should be **robust** to noisy and low-quality audio. Upon previous SSL-based VC methods, we enable these properties of our model with several improvements:

**Multilingual HuBERT.** To enable the model to be cross-lingual, we employ a multilingual HuBERT (Lee et al., 2021; Popuri et al., 2022) to extract the SSL discrete representations as the content information, which is pre-trained on speech in multiple languages. We find it also generalizes well to other unseen languages (see Appendix D.2). HuBERT does not need any paired data or speaker

information, which is consistent with our task setting. Besides, we add a language ID input to the speech decoder to indicate the language we need to generate, which can accurately model the pronunciation differences among different languages given the same discrete representation, and make up for the limited capacity of multilingual representations.

**Variational encoder with flow-based enhanced prior.** To improve model robustness and audio quality, inspired by previous successful work in TTS (Ren et al., 2021b; Kim et al., 2021a), we introduce a variational encoder with flow-based enhanced prior. In training, this encoder can "store" the residual information, *e.g.*, some irregular noises, time-varying timbre and prosody, that cannot be encoded by other information extraction modules to the posterior distribution $D_q$, and uses a normalizing flow to reshape the prior distribution $D_p$ which needs to be close to $D_q$ in terms of KL-divergence. With normalizing flows, the KL-divergence no longer offers a simple closed-form solution. So we estimate KL-divergence via Monte-Carlo method as in Ren et al. (2021b) and Kim et al. (2021a). The reason why we need the normalizing flow is that simple Gaussian prior distribution results in strong constraints on the posterior, which pushes the posterior distribution towards the mean and limits diversity, while the distribution shaped by normalizing flows is more flexible and provides the decoder with stronger prior. Besides, it can also provide the sampled random variables with temporal dependency.

**Information bottleneck.** HuBERT representation is a kind of low-bitrate representation for speech content and does not contain much non-lexical information such as speaker identity and emotion. However, due to its discrete space bottleneck and way of training, we still cannot ensure it fully disentangles the timbre information, and the remaining timbre information may degrade the voice conversion quality in the inference stage. To further erase the speaker identity information, we need to choose an appropriate input dimension for the content encoder (*i.e.*, embedding layer for HuBERT tokens), which can neither be too large nor too small. A large dimension may lead to leakage of fine-grained identity information from the content encoder and a small one may result in loss of pronunciation information. We put more details of our VC model in Appendix B.1.

### 3.3 TTS AND ASR MODELS

Previous unsupervised TTS works use an autoregressive (AR) TTS architecture such as Tacotron 2 (Shen et al., 2018) and TransformerTTS (Li et al., 2018), which automatically find the speech-text alignment. However, such an AR TTS architecture is not robust and prone to word missing and repeating problems as stated in Ren et al. (2019a). In this work, as shown in Figure 2b, we adopt a non-autoregressive (NAR) TTS architecture (Ren et al., 2019a; 2021a). We mainly follow PortaSpeech (Ren et al., 2021b), except that we replace the post-net in PortaSpeech with multi-length adversarial training (Ye et al., 2022; Chen et al., 2020) to simplify the training pipeline while keeping the naturalness of the generated mel-spectrogram. Instead of obtaining the ground-truth duration information from Montreal Forced Aligner (MFA) (McAuliffe et al., 2017) as many non-autoregressive TTS models (Ren et al., 2021a;b; Ye et al., 2022) do, we extract the speech-text alignment from the attention matrix generated by the ASR model, which simplifies the training pipeline in our back-translation stage and removes the dependency upon external tools. Specifically, inspired by GlowTTS (Kim et al., 2020), we extract the speech-text alignment by finding the monotonic path of maximum probability over the attention matrix of our ASR model using the Viterbi decoding. To enable the TTS model to generate speech in a different language, we add a language embedding to the decoder and an extra language ID input is required to specify the language of the target speech.

Our ASR model is based on a sequence-to-sequence architecture with an LSTM-based encoder and decoder. To generate more monotonic alignment for TTS training, we employ the location-sensitive attention (Shen et al., 2018). Different from the TTS model, our ASR model is universal to all languages and does not need any language embedding as input, which can generalize to new languages better in our scenario. We put more details and model configurations of TTS and ASR models in Appendix B.2 and B.3.

### 3.4 TRICKS IN BACK-TRANSLATION

Back-translation is a very critical step for unsupervised TTS training to leverage unpaired speech and text data. In this subsection, we describe some back-translation strategies that can significantly improve the effectiveness and efficiency of unsupervised TTS training.

**Curriculum learning.** After warming up the ASR model in Stage 2 using $S_{rich}$ and $T_{rich}$, we can force the ASR model to transcribe the audio in $L_{low}$ to the text in $L_{rich}$ by initializing the language embedding of $L_{low}$ with that of $L_{rich}$. Considering the results of ASR are taken as the input of TTS, we select some good transcriptions, whose pronunciation is very similar to the ground-truth, for TTS training and discard bad ones in each round of back-translation. Apparently, we cannot directly calculate the error rate between the transcription and ground-truth text since we have no corresponding text for each audio. Therefore, to filter good recognition results during iterative back-translation, we design a metric called *focus rate* ($\mathcal{F}$) to evaluate the confidence of ASR results, which is defined as $\mathcal{F} = \frac{1}{N} \sum_{i=1}^{N} A_{i,P_i}$. In its definition, $N$ denotes the number of mel-spectrogram frames; $A_{i,j}$ is ASR attention weights at the position of the $i$-th mel-spectrogram frame and the $j$-th text token and satisfies $\sum A_i = 1$; $P_i$ is the text token index corresponding to the $i$-th mel-spectrogram frame in the monotonic path of maximum probability decoded by the Viterbi algorithm (Forney, 1973). A higher $\mathcal{F}$ means greater probability lies in the decoded monotonic path and implies better speech-text monotonic alignment and higher confidence in the transcribed results. In each round of the pseudo text generation process, we use $\mathcal{F}$ to select good ASR results whose $\mathcal{F}$ is greater than a fixed threshold $\mathcal{F}_{thres}$. Besides, we also store $\mathcal{F}$ for each pseudo text $t_{\text{pseudo}}$ and replace $t_{\text{pseudo}}$ with new result in the next back-translation round only if $\mathcal{F}$ is increased.

**Length augmentation.** At the beginning of training in the low-resource language, short utterances (text and speech) are easier for TTS and ASR models to fit and they are generally better at generating short utterances rather than long ones. Besides, our curriculum learning strategy approximates the quality of the generated text and filters bad results, forcing our model to keep more short utterances than long ones. Consequently, our model becomes biased towards short utterances and may perform very poorly for long sentences. To fix this issue, we introduce a length augmentation strategy. In particular, we randomly concatenate two utterances $(t^1, t^2)/(s^1, s^2)$ and their generated results $(s^1_{\text{pseudo}}, s^2_{\text{pseudo}})/(t^1_{\text{pseudo}}, t^2_{\text{pseudo}})$ with some probability $p_{\text{cat}}$ and obtain the generated pairs $(t^{\text{cat}}, s^{\text{cat}}_{\text{pseudo}})$ and $(s^{\text{cat}}, t^{\text{cat}}_{\text{pseudo}})$ for back-translation training. Length augmentation helps the TTS and ASR models generate long sentences better and become more robust to some long text inputs in inference.

**Auxiliary supervised losses.** If we only employ the back-translation loss in Stage 3, the model may fail to find the correct speech-text alignment, leading to unsatisfied results and unstable training, especially at the beginning of training. To solve this problem, apart from the back-translation loss in the target low-resource language, we also keep the supervised training losses in the auxiliary rich-resource language the same as those in Stage 2. We call them "auxiliary supervised losses". Specifically, in the process of training, we intersperse the rich resource language supervised training, for both TTS and ASR, into the back-translation steps with some probability $p_{aux}$.

## 4 EXPERIMENTS AND RESULTS

In this section, we conduct experiments to evaluate the effectiveness of our proposed method for unsupervised TTS. We first describe the experiment settings, show the results of our method, and conduct some analyses of our method.

### 4.1 EXPERIMENTAL SETUP

**Datasets.** We choose the speech and text data from CommonVoice dataset (Ardila et al., 2019) for training and English and Indonesian as the target low-resource languages[4]. We use French as the rich-resource language unless otherwise stated. The experimental results of using other languages as the rich-resource language are put in Appendix D.2. We split the target language data into two halves. We take unpaired speech data from the first half and text data from the second, so as to guarantee the speech and text data are disjoint. We use LJSpeech (Ito, 2017) as the $S_{ref}$ to provide the speaker timbre and suppress the background noise for the voice conversion model. For evaluation, we choose 100 audio/text pairs in LJSpeech for English[5] and 100 audio/text pairs in CommonVoice (Indonesian subset) for Indonesian. For the speech data, we convert the raw waveform into mel-

---

[4]We choose English as one of the target languages since we can understand English and it is easy to evaluate, although English is not a low-resource language.

[5]We use LJSpeech because it has fewer errors in text and speech pairing, while the data in CommonVoice are very noisy and have much wrongly labeled text.

Table 2: The comparison between our method and other existing unsupervised TTS methods. Liu et al. (2022b) and Ni et al. (2022) are same in our settings.

| Methods | English | | | Indonesian | | |
|---|---|---|---|---|---|---|
| | MOS | CER | WER | MOS | CER | WER |
| Supervised | 4.11±0.07 | 0.016 | 0.052 | 4.07±0.08 | 0.024 | 0.068 |
| Ren et al. (2019b) | 3.32±0.12 | 0.393 | 0.684 | 3.25±0.11 | 0.392 | 0.702 |
| Xu et al. (2020) | 3.42±0.10 | 0.376 | 0.645 | 3.39±0.13 | 0.389 | 0.695 |
| Liu et al. (2022b); Ni et al. (2022) | 3.49±0.11 | 0.305 | 0.555 | 3.52±0.09 | 0.299 | 0.536 |
| Ours | 3.82±0.09 | 0.145 | 0.320 | 3.98±0.10 | 0.034 | 0.083 |

spectrograms with 80 ms frame size, 20 ms frame hop following Hsu et al. (2021). More details are listed in Appendix C.1.

**Training and evaluation.** We train our VC, TTS, and ASR models on 1 NVIDIA A100 GPU witch batch size 128. We use the Adam optimizer with $\beta_1 = 0.9$, $\beta_2 = 0.98$, $\varepsilon = 10^{-9}$ and learning rate 2e-4. The training takes nearly 3 days. The output mel-spectrograms are converted to waveform using a HiFi-GAN (Kong et al., 2020) pre-trained on LJSpeech (Ito, 2017). The focus rate $\mathcal{F}_{thres}$, $N_{steps}$, $p_{cat}$ and $p_{aux}$ in back-translation are set to 0.2, 20k, 0.2 and 0.2. For evaluation, we mainly use MOS (mean opinion score) for audio quality, WER (word error rate), and CER (character error rate) for the intelligibility of the voice (French & Steinberg, 1947) to verify if we can generate a reasonable speech sequence. For mean opinion score evaluation, we keep the text content consistent among different models so as to exclude other interference factors and only examine the audio quality. We randomly choose 20 sentences from the test set and each audio is listened by at least 20 testers following Ren et al. (2019a; 2021a), who are all native English/Indonesian speakers. For WER and CER, we first transcribe the sentences from the generated speech using open-sourced or commercial ASR and calculate these metrics between them and the ground-truth text in the test set. We use WeNet (Yao et al., 2021; Zhang et al., 2022) for English for fair comparison with future works, since commercial ASR could be changed in the future; but we choose Azure ASR service[6] for Indonesian since we cannot find any Indonesian open-sourced ASR that is accurate enough. In analytical experiments, we also show the CER of our ASR model, which also indicates the performance of our system since ASR and TTS are dependent on each other and boosted iteratively.

## 4.2 RESULTS AND ANALYSES

### 4.2.1 PERFORMANCE

We compare our method with previous works including Ren et al. (2019b), Xu et al. (2020), Liu et al. (2022b) and Ni et al. (2022). For fair comparison, we make some modifications to all baseline methods including unifying the training dataset, TTS acoustic model and vocoder (more detailed modifications of each baseline method are put in Appendix C.2). The results are shown in Table 2. We also evaluate the outputs of a supervised TTS model trained with paired target language data for reference, and its results can be regarded as the upper bound. From the table, it can be seen that our method achieves the best performance in both speech quality (MOS) and intelligibility (CER and WER) in English and Indonesian. And very surprisingly, our method can even approach the performance of the supervised model in Indonesian. A possible reason is that Indonesian is easier to pronounce than English. These observations prove the effectiveness of our proposed tricks for unsupervised TTS.

### 4.2.2 ABLATION STUDY

To analyze the effectiveness of each trick and component, we conduct some ablation studies on English. In addition to generated speech quality (MOS) and intelligibility (CER and WER), we also analyze the character error rate of our ASR model (CER(ASR)). The results are shown in Table 3. 1) From Row 2, it can be seen that our model can achieve better performance after normalizing the speech variance including timbre and noise in our dataset.

---

[6] https://azure.microsoft.com/en-us/products/cognitive-services/speech-to-text/

2) From Row 3, it can be seen that the NAR TTS architecture improves speech quality and intelligibility by a large margin, as NAR TTS is more robust to noisy speech and reduces some bad cases in generated speech. 3) From Row 4, it can be seen that back-translation is essential to unsupervised TTS, which is consistent with the findings of previous works (Xu et al., 2020; Ren et al., 2019b). 4) From Row 5, we can see that curriculum learning can improve the training effectiveness since it can filter out

Table 3: Ablation studies on the components of our method. "Norm" denotes normalization via voice conversion model; "NAR" denotes non-autoregressive TTS architecture; "CL" denotes curriculum learning; "Aug" denotes the length augmentation; "Aux" denotes the auxiliary supervised training loss; "BT" denotes back-translation.

| No. | Settings | MOS | CER | WER | CER (ASR) |
|---|---|---|---|---|---|
| 1 | Ours | 3.82±0.09 | 0.145 | 0.320 | 0.445 |
| 2 | w/o. Norm | 3.21±0.12 | 0.431 | 0.789 | 0.684 |
| 3 | w/o. NAR | 3.18±0.09 | 0.496 | 0.756 | 0.627 |
| 4 | w/o. BT | 3.39±0.10 | 0.376 | 0.706 | / |
| 5 | w/o. CL | 3.72±0.08 | 0.225 | 0.493 | 0.481 |
| 6 | w/o. Aug | 3.73±0.09 | 0.163 | 0.358 | 0.479 |
| 7 | w/o. Aux | 3.70±0.09 | 0.227 | 0.472 | 0.575 |

bad pseudo transcripts and improve the training set quality for back-translation. 5) From Row 6, it can be seen that our length augmentation strategy can improve the robustness to long text inputs in inference. 6) From Row 7, we find that auxiliary supervised training loss can improve the performance of both ASR and TTS by stabilizing the training. From the table, comparing other rows with Row 1 that shows our model with all tricks, we have several observations.

### 4.2.3 Analyses on Voice Conversion Model

With verified effectiveness of normalization via the voice conversion model as demonstrated in Section 4.2.2, we conduct more analyses on our proposed voice conversion model, including the effects of different information bottleneck channels and the flow-based enhanced prior. The results are shown in Table 4. It can be observed that an appropriate size of bottleneck channels is crucial for the performance of the voice conversion

Table 4: The comparison between our unsupervised TTS with others whose training speech is normalized with different voice conversion methods. "Var. Enc." represents our variational encoder with flow-based enhanced prior; "Chn" denotes the content information bottleneck channels.

| No. | Var. Enc. | Chn. | MOS | CER | WER | CER (ASR) |
|---|---|---|---|---|---|---|
| 1 | ✓ | 16 | 3.82±0.09 | 0.145 | 0.320 | 0.445 |
| 2 | ✗ | 16 | 3.75±0.10 | 0.155 | 0.351 | 0.500 |
| 3 | ✓ | 8 | 3.69±0.10 | 0.191 | 0.384 | 0.496 |
| 4 | ✓ | 32 | 3.79±0.09 | 0.146 | 0.342 | 0.478 |
| 5 | ✓ | 128 | 3.70±0.11 | 0.199 | 0.396 | 0.507 |

model, with a large bottleneck resulting in timbre information leakage and a small bottleneck leading to pronunciation information loss. Besides, our flow-based enhanced prior can improve the quality of converted speech, since it has make fewer assumptions about the prior distribution as we mentioned in Section 3.2.

## 5 Conclusion

In this work, we proposed an unsupervised method for TTS by leveraging low-quality and noisy unpaired speech and text data in the target language and paired data in other rich-resource languages. Our method encloses several practical tricks to realize unsupervised text-to-speech, including normalizing variable and noisy information in speech data, curriculum learning, length augmentation, and auxiliary supervised training. We have also found that the non-autoregressive TTS architecture can significantly relieve robustness issues in unsupervised settings. We conducted experiments on CommonVoice dataset, taking English and Indonesian as the target languages, and have found that our method can achieve high audio quality in terms of MOS, and high intelligibility in terms of WER and CER, demonstrating remarkable effectiveness. Further analyses have well verified the importance of each trick of our method.

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

# Appendices

## A    TRAINING ALGORITHM

The detailed unsupervised training algorithm is shown in Algorithm 1.

---

**Algorithm 1** Unsupervised TTS Training

---

1: **Input**: paired dataset in rich-resource language $S_{\text{rich}}$ and $T_{\text{rich}}$; unpaired speech and text data in low-resource language $S_{\text{low}}$ and $T_{\text{low}}$; single-speaker speech dataset $S_{\text{ref}}$ containing reference speaker for voice conversion; pre-trained multilingual HuBERT model $M_{\text{h}}$; pre-trained speaker encoder $M_{\text{spk}}$.
2: **Initialize**: multilingual TTS model $M_{\text{TTS}}$ and ASR model $M_{\text{ASR}}$; current unsupervised training step $t = 0$; total unsupervised training steps $T_{total}$; number of steps for each TTS or ASR stage $N_{step}$.
3: Train our proposed voice conversion model with $S_{\text{rich}}$, $S_{\text{low}}$ and $S_{ref}$, and use the $M_{\text{h}}$ and $M_{\text{spk}}$ to extract HuBERT and speaker representations.
4: Convert the timbre of all speech samples in $S_{\text{rich}}$ and $S_{\text{low}}$ to that of the speech in $S_{\text{ref}}$ and obtain the converted $S'_{\text{rich}}$ and $S'_{\text{low}}$.                                             {Sec. 3.2}
5: Train $M_{ASR}$ and $M_{TTS}$ using $S'_{\text{rich}}$ and $T_{\text{rich}}$.
6: **repeat**
7:     Convert all samples $s$ in $S'_{\text{low}}$ to pseudo text $t_{\text{pseudo}}$.
8:     Select pseudo training pairs $(t_{\text{pseudo}}, s)$ satisfying $\mathcal{F} > \mathcal{F}_{\text{thres}}$ and
            obtain $(T_{\text{pseudo}}, S')$.                                          {Curriculum learning in Sec. 3.4}
9:     **for** $N$ in 0 to $N_{\text{steps}}$ **do**
10:        **if** Random() $\leq p_{aux}$ **then**
11:            Sample $D \leftarrow (t_{\text{rich}}, s)$ from $(T_{\text{rich}}, S'_{\text{rich}})$.                         {Auxiliary loss in Sec. 3.4}
12:        **else**
13:            Sample $D \leftarrow (t_{\text{pseudo}}, s)$ from $(T_{\text{pseudo}}, S')$.
14:            **if** Random() $\leq p_{cat}$ **then**
15:                Sample $D' \leftarrow (t^2_{\text{pseudo}}, s^2)$ from $(T_{\text{pseudo}}, S')$.
16:                $D \leftarrow (Concat(t_{\text{pseudo}}, t^2_{\text{pseudo}}), Concat(s, s^2))$         {Length augmentation in Sec. 3.4}
17:            **end if**
18:        **end if**
19:        Train $M_{TTS}$ using $D$.
20:     **end for**
21:     **for** $N$ in 0 to $N_{\text{steps}}$ **do**
22:        Convert all samples $t$ in $T_{low}$ to pseudo speech $s_{\text{pseudo}}$ and obtain $(S_{\text{pseudo}}, T_{low})$.
23:        **if** Random() $\leq p_{aux}$ **then**
24:            Sample $D \leftarrow (s_{\text{rich}}, t)$ from $(S'_{\text{rich}}, T_{\text{rich}})$.                         {Auxiliary loss in Sec. 3.4}
25:        **else**
26:            Sample $D \leftarrow (s_{\text{pseudo}}, t)$ from $(S_{\text{pseudo}}, T_{low})$.
27:            **if** Random() $\leq p_{cat}$ **then**
28:                Sample $D' \leftarrow (s^2_{\text{pseudo}}, t^2)$ from $(S_{\text{pseudo}}, T_{low})$.
29:                $D \leftarrow (Concat(s_{\text{pseudo}}, s^2_{\text{pseudo}}), Concat(t, t^2))$         {Length augmentation in Sec. 3.4}
30:            **end if**
31:        **end if**
32:        Train $M_{ASR}$ using $D$.
33:     **end for**
34:     $t \leftarrow t + 1$
35: **until** $t > T_{total}$

---

## B    MODEL DETAILS AND CONFIGURATIONS

In this section, we put more details of models including voice conversion (VC), text-to-speech (TTS), and automatic speech recognition (ASR) models, and also the hyper-parameters we used in our experiments.

### B.1    VC MODEL

Our proposed VC model takes two mel-spectrograms (content provider $M_1$ and timbre reference $M_2$) as inputs and outputs the converted mel-spectrogram $M_3$. Firstly, $M_1$ is fed into a pre-trained

speaker encoder[7] to extract the speaker embedding $H_{spk}$. Secondly, $M_2$ is fed into a pre-trained *multilingual HuBERT*[8], which is pre-trained with three languages, and extract the HuBERT discrete frame-level representation $H_{ling}$. Thirdly, $M_2$ is taken to the *posterior encoder*, which generates a multivariate Gaussian distribution as the posterior in our variational VC model. Instead of directly employing Gaussian distribution, we introduce a small *volume-preserving normalizing flow* to model the prior distribution. A latent $\mathbf{z}$ is sampled from the posterior distribution (in training) or prior distribution (in inference). Finally, we add $H_{spk}$, $H_{ling}$, $\mathbf{z}$ and language embedding of $M_2$ together (all of them have the same channel size $C = 192$) and feed the result hidden states into the speech decoder to generate the target speech. Besides, we introduce a *multi-length discriminator* to distinguish between the output generated by the model and the ground truth mel-spectrogram.

The loss terms of the voice conversion model consist of 1) reconstruction loss of mel-spectrotram $L_{MAE}$: mean absolute error between the generated and ground-truth mel-spectogram; 2) the KL-divergence of prior and posterior distributions: $L_{KL} = \log q_\phi(\mathbf{z}|\mathbf{x}) - \log p_{\bar{\theta}}(\mathbf{z})$, where $\mathbf{z} \sim q_\phi(\mathbf{z}|\mathbf{x})$; and 3) the adversarial training loss introduced by the *multi-length discriminator*: $L_{adv}$. The final weighted total loss is $L_{total} = \lambda_1 L_{MAE} + \lambda_2 L_{KL} + \lambda_3 L_{adv}$. In our experiments, we set $\lambda_1 = \lambda_2 = \lambda_3 = 1.0$.

The detailed structure of each module is introduced in the following subsubsections.

### B.1.1 MULTILINGUAL HuBERT

Multilingual HuBERT (Lee et al., 2021) is trained on English (En), Spanish (Es), and French (Fr) 100k subsets of the VoxPopuli dataset (Wang et al., 2021). VoxPopuli contains unlabeled speech data for 23 languages, and Lee et al. (2021) use the 4.5k hrs of unlabeled speech for En, Es, and Fr, totaling 13.5k hours. We extract the HuBERT features from the 11-th layer of the third-iteration HuBERT model and discretize them using the pre-trained K-means model to obtain the discrete representations $H_{ling}$.

### B.1.2 POSTERIOR ENCODER

The structure of posterior encoder is similar with the encoder in the variational generator of PortaSpeech (Ren et al., 2021b), which is composed of a 1D-convolution with stride 4 followed by ReLU activation (Glorot et al., 2011) and layer normalization (Ba et al., 2016), and a non-causal WaveNet (Van Den Oord et al., 2016), as shown in Figure 3a. The number of encoder layers, WaveNet channel size and kernel size are 8, 192 and 5. The outputs of posterior encoder is the parameters ($\mu_q$ and $\sigma_q$) of the posterior distribution $N(\mu_q, \sigma_q)$ and the latent $z$ is sampled from $N(\mu_q, \sigma_q)$, whose latent size is set to 32.

### B.1.3 VOLUME-PRESERVING (VP) NORMALIZING FLOW

Following Kim et al. (2021b) and Ren et al. (2021b), we use volume-preserving normalizing flow as the prior distribution generator since it does not need to consider the Jacobian term when calculating the data log-likelihood and is powerful enough for modeling the prior, as shown in Figure 3c. The normalizing flow transforms simple distributions (*e.g.*, Gaussian distribution) to complex distributions through a series of K invertible mappings, which is a stack of WaveNet (van den Oord et al.) residual blocks with dilation 1. Then we take the complex distributions as the prior of the speech decoder. When introducing normalizing flow-based enhanced prior, the optimization objective of the mel-spectrogram generator becomes:

$$\log p(\mathbf{x}) \geq \mathbb{E}_{q_\phi(\mathbf{z}|\mathbf{x})}[\log p_\theta(\mathbf{x}|\mathbf{z})] - \mathrm{KL}(q_\phi(\mathbf{z}|\mathbf{x})|p_{\bar{\theta}}(\mathbf{z})) \equiv \mathcal{L}(\phi, \theta, \bar{\theta}), \quad (1)$$

where $\phi$, $\theta$ and $\bar{\theta}$ denote the model parameters of the posterior encoder, speech decoder and the normalizing flow-based enhanced prior, respectively. Due to the introduction of normalizing flows, the KL term in Equation 1 no longer offers a simple closed-form solution. So we estimate the expectation w.r.t. $q_\phi(\mathbf{z}|\mathbf{x})$ via Monte-Carlo method by modifying the KL term:

$$\mathrm{KL}(q_\phi(\mathbf{z}|\mathbf{x})|p_{\bar{\theta}}(\mathbf{z})) = \mathbb{E}_{q_\phi(\mathbf{z}|\mathbf{x})}[\log q_\phi(\mathbf{z}|\mathbf{x}) - \log p_{\bar{\theta}}(\mathbf{z})]. \quad (2)$$

---

[7]https://github.com/resemble-ai/Resemblyzer

[8]https://github.com/facebookresearch/fairseq/blob/main/examples/speech_to_speech/docs/textless_s2st_real_data.md

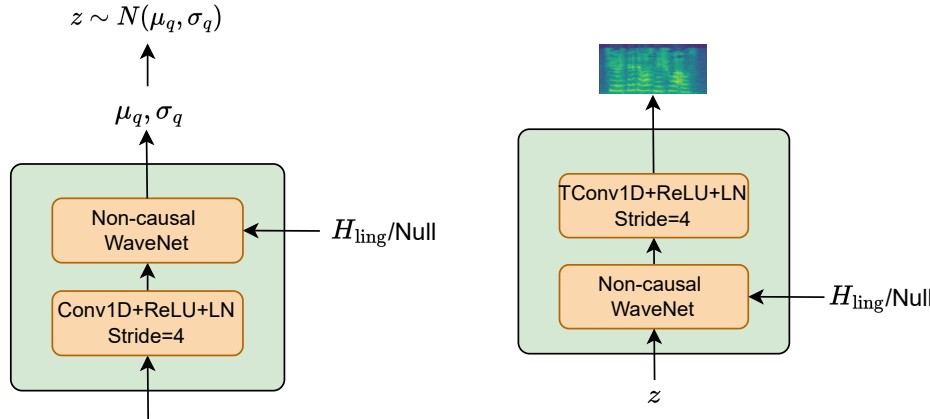

(a) Posterior encoder. In the voice conversion model, the condition is "Null". In TTS model, the condition is linguistic feature $H_{\text{ling}}$.

(b) Speech decoder.

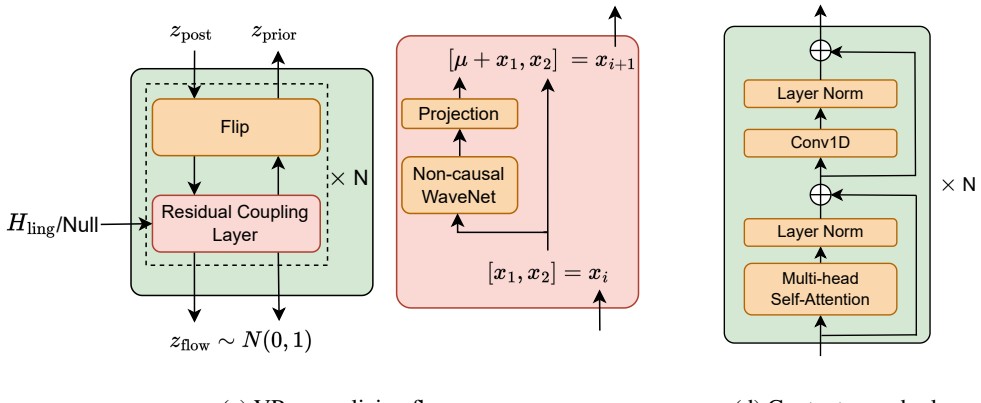

(c) VP normalizing flow.

(d) Content encoder layers.

Figure 3: Modules in our voice conversion model.

In training, the posterior distribution $N(\mu_q, \sigma_q)$ is encoded by the encoder of the posterior encoder. Then **z** is sampled from the posterior distribution using reparameterization and is passed to the speech decoder. In the meanwhile, the posterior distribution is fed into the VP normalizing flow to convert it to a standard normal distribution (the middle dotted line). In inference, VP normalizing flow converts a sample in the standard normal distribution into a sample **z** and we pass the **z** to the speech decoder.

Our VP normalizing flow consists of 4 flow steps, each of which has 4 WaveNet layers, whose channel size and kernel size are set to 64 and 3.

### B.1.4 CONTENT ENCODER

The content encoder is stacks of feed-forward Transformer (Vaswani et al., 2017) layers with relative position encoding (Shaw et al., 2018), as shown in Figure 3d. The information bottleneck is located in the first layer of the content encoder (the embedding layer of HuBERT tokens). We set the channel size of each embedding to 16 as default.

### B.1.5 SPEECH DECODER

The speech decoder, as shown in Figure 3b, consists of a non-causal WaveNet and a 1D transposed convolution with stride 4, also followed by ReLU and layer normalization. The number of decoder layers, WaveNet channel size and kernel size are set to 4, 192 and 5.

### B.1.6    MULTI-LENGTH DISCRIMINATOR

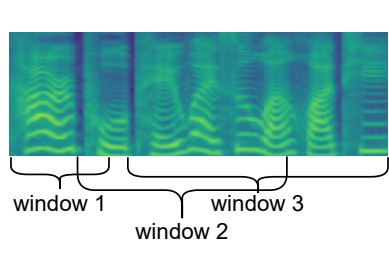

(a) Multi-window clips in mel-spectrogram.

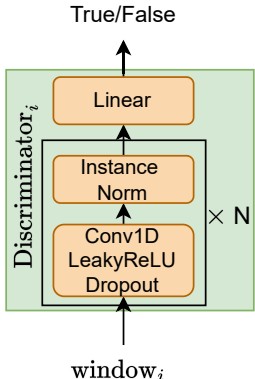

(b) Discriminator structure.

Figure 4: Multi-length Discriminator.

Inspired by Ye et al. (2022), our multi-length discriminator is an ensemble of multiple CNN-based discriminators, which evaluate the mel-spectrogram based on random windows of different lengths, as shown in Figure 4. In our experiments, we train three CNN-based discriminators which observe random mel-spectrogram clips with lengths of 32, 64, and 128 frames. The structure of the CNN-based discriminator is shown in Figure 4b. It consists of N+1 2D-convolutional layers, each of which is followed by a Leaky ReLU activation and drop-out layer. The latter N convolutional layers are additionally followed by an instance normalization (Ulyanov et al., 2016) layer. After the convolutional layers, a linear layer projects the hidden states of the mel-spectrogram slice to a scalar, which is the prediction that the input mel-spectrogram is true or fake. In our experiments, we set N=2 and the channel size of these discriminators to 32.

### B.2    TTS MODEL

Our TTS model architecture follows PortaSpeech (Ren et al., 2021b) except that 1) we replace the post-net in PortaSpeech with multi- length adversarial training, which is the same as the adversarial training in our voice conversion model in Appendix B.1. 2) We add a language embedding layer to the speech decoder, indicating the language of the speech that will be generated. 3) We use a simple character encoder like FastSpeech (Ren et al., 2019a; 2021a) instead of the mixed linguistic encoders for simplicity. The detailed model architecture and hyper-parameters of *posterior encoder, normalizing flow, speech decoder and multi-length discriminators* are the same as those in the voice conversion model in Appendix B.1. The structures of *text encoder* and *duration predictor* are the same as those in Ren et al. (2021b), with channel size 192, kernel size 5 and number of layers 4.

### B.3    ASR MODEL

We adopt the architecture of Tacotron 2 (Shen et al., 2018) for our ASR model, since its location-sensitive attention can generate close-to-diagonal and monotonic alignment between speech and text. We replace the character/phoneme embedding of the encoder in Tacotron 2 with a speech CNN-based pre-net with stride 4 to enable speech information encoding. For the decoder side, we use the character embedding as the input layer and the softmax as the output layer to adapt the decoder to the character sequence. We set the hidden size of encoder RNN to 512 and the number of convolution stacks to 5; the hidden size of decoder RNN and attention RNN are both set to 1024; the channel of decoder attention is set to 512. We also train a Transformer-based (Vaswani et al., 2017) language model and jointly beam search decode the recognition results $X$ using the ASR model $p_{\text{ASR}}(X|S)$ and language model $p_{\text{LM}}(X)$ to maximize the probability $logp_{\text{ASR}}(X|S) + \lambda_{\text{LM}}logp_{\text{LM}}(X)$, where $\lambda_{\text{LM}}$ is the weight of language model and it is set to 0.2 in this work. The beam size of beam search decoding is set to 5.

## C  MORE EXPERIMENTAL DETAILS

In this section, we describe more experimental details for reproducibility.

### C.1  DATASETS

We select subsets of English, French and Indonesian from CommonVoice (Ardila et al., 2019) dataset. We choose subsets of about 200k utterances for English and French and all data (about 20k utterances) for Indonesian. We randomly select 100 utterances in English and Indonesian for validation and another 100 utterances in Indonesian for testing. The test set of English is randomly selected from LJSpeech (Ito, 2017). We split the target language data into two halves according to utterance ID: we take unpaired speech data from those with odd ID and text data from those with even ID, so as to guarantee the speech and text data are disjoint.

### C.2  BASELINES

For fair comparison, we make some modifications to all baseline methods as follows:

- We adopt training data consisting of paired French (as auxiliary rich-resource language) data and unpaired English/Indonesian (as target low-resource language) data. Specifically, for Ren et al. (2019b) and Xu et al. (2020), we warmup ASR and TTS in these methods using rich-resource language data before back-translation; for Liu et al. (2022b) and Ni et al. (2022), we initialize the unsupervised ASR model using a modified CTC loss (Graves et al., 2006) with rich-resource language data before unsupervised training and also initialize the TTS model using this data.
- We directly use character sequence as TTS input without any lexicon and G2P tools.
- We extract the speaker embeddings using the same pre-trained speaker encoder[9] and add them to the TTS model to indicate the speaker information (timbre) since our dataset is multi-speaker.
- We replace all baseline TTS models with NAR architecture the same as our method since AR architecture is very sensitive to noisy audio and cannot produce any meaningful results in our settings.
- We use the same voice conversion model (described in Section 3.2) to convert ground-truth audios and all baselines' outputs to the same person's timbre from $S_{ref}$.
- We use the same vocoder, HiFi-GAN (Kong et al., 2020), to convert mel-spectrogram to the waveform.

## D  MORE EXPERIMENTAL RESULTS

### D.1  PERFORMANCE CHANGES IN BACK-TRANSLATION TRAINING

To verify the accuracy of ASR and the performance of TTS can be boosted iteratively as the training directions shift, we plot the accuracy of TTS and ASR results in Figure 5. From the figure, we can see that with the training of the model (the training directions shift every $N_{steps} = 20000$ steps), the error rates of ASR and TTS results gradually drop until convergence.

### D.2  USE OTHER RICH-RESOURCE AUXILIARY LANGUAGES

We explore how different rich-resource auxiliary languages can affect the target language's performance. In addition to French, we use other languages including German, Dutch, Spanish, and Portuguese as the rich-resource auxiliary language to train our unsupervised TTS model. For fair comparison, we use the same training data size for all rich-resource languages (80k pairs speech-text subset in each language from CommonVoice). We choose English as the target low-resource language. The results are shown in Table 5. It can be seen that using German as the rich-resource

---

[9]https://github.com/resemble-ai/Resemblyzer

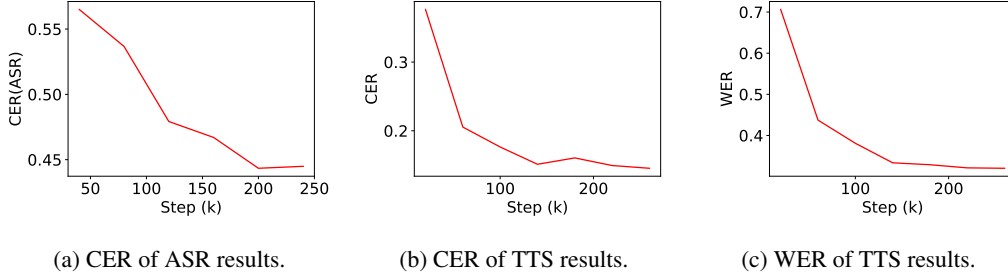

(a) CER of ASR results.  (b) CER of TTS results.  (c) WER of TTS results.

Figure 5: Accuracy of unsupervised English TTS and ASR models.

language achieves the best performance. The possible reason is that the pronunciation distance between English and German is closer than other languages as they both belong to west Germanic languages. Though Dutch also belongs to west Germanic languages, it does not perform very well, which might be due to its bad data quality. Then we combine data from all these languages and find that it achieves very strong results and outperforms others that use only one auxiliary language. Besides, we observe that our method performs very well not only in English, French and Spanish, which are used to pre-trained the multilingual HuBERT, but also in other unseen languages, which verifies the generalization of our voice conversion model and the whole unsupervised TTS pipeline.

Table 5: Performances of unsupervised English TTS and ASR models which are initialized with supervised training in different rich-resource auxiliary languages.

| Auxiliary languages | CER | WER | CER (ASR) |
|---|---|---|---|
| French | 0.195 | 0.377 | 0.475 |
| German | 0.113 | 0.264 | 0.370 |
| Dutch | 0.257 | 0.522 | 0.547 |
| Spanish | 0.207 | 0.423 | 0.539 |
| Portuguese | 0.225 | 0.468 | 0.540 |
| All | 0.068 | 0.168 | 0.288 |

### D.3 ANALYSES ON FOCUS RATE $\mathcal{F}$

To verify the effectiveness of focus rate $\mathcal{F}$ we propose in Section 3.4, we calculate $\mathcal{F}$ and CER on English test set in our model training process. We plot the curves to explore the correlation between them in Figure 6. From the figure we can see that the focus rate $\mathcal{F}$ is negatively related to recognition accuracy, which means it is reasonable to use it as the indicator for filtering ASR transcriptions (higher $\mathcal{F}$ indicates lower CER).

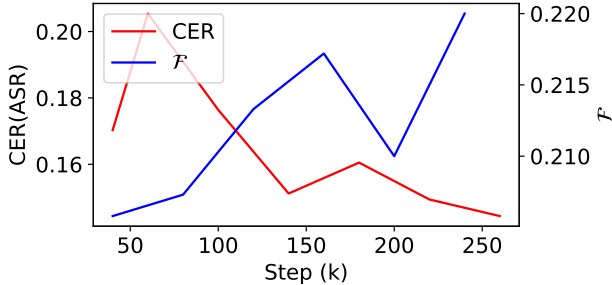

Figure 6: $\mathcal{F}$ and CER evaluated on English test set in the training process.

Table 6: The results of our method trained with different text unpaired dataset.

| Dataset | MOS | CER | WER |
|---|---|---|---|
| Supervised | 4.11±0.07 | 0.016 | 0.052 |
| Ours (CommonVoice text) | 3.82±0.09 | 0.145 | 0.320 |
| Ours (WMT text) | 3.74±0.08 | 0.181 | 0.380 |

### D.4 USE OTHER TEXT UNPAIRED DATASET

We train our model using the audio data from the CommonVoice English subset which is the same as the original version of the paper and the text data from WMT16 (Bojar et al., 2016) English training set to make the domains of unpaired audio and text very different. We keep the test set the same as the original paper (LJSpeech subset). The results are shown in Table 6. From the table, it can be seen that the performance drops a bit ($\sim$0.036 and $\sim$0.06 increasing in CER and WER) due to the domain gap between the text and speech unpaired data.

### E POTENTIAL NEGATIVE SOCIETAL IMPACTS

Unsupervised TTS lowers the requirements for speech synthesis service deployment (only needs unpaired speech and text data) and synthesizes high-quality speech voice, which may cause unemployment for people with related occupations such as broadcasters and radio hosts. In addition, there is the potential for harm from non-consensual voice cloning or the generation of fake media and the voices of the speakers in the recordings might be overused than they expect.

