# OpenReview forum: "Bag of Tricks for Unsupervised Text-to-Speech"
_ICLR.cc/2023/Conference — ICLR 2023 poster_

### Official Review · Reviewer_sG8y · 2022-10-24

**Confidence:** 3
**Correctness:** 3
**Technical Novelty And Significance:** 3
**Empirical Novelty And Significance:** 3
**Recommendation:** 8

**Clarity, Quality, Novelty And Reproducibility:**

The contributions are somewhat new. Aspects of the contributions exist in prior work.


**Strength And Weaknesses:**

Strength:

The main contributions are clearly stated and supported by the experiments.

Major works on the similar topic are widely covered and referenced.

The samples are quite convincing.

Evaluation is thorough enough to support the arguments with in-depth ablation studies.

Weaknesses:

The complicated training algorithm makes the proposed method hard to reproduce. It would be helpful if the authors can provide brief guidelines such as how to tune the hyper parameters. It seems that the performance of this system heavily depends on the performance of the self-supervised voice conversion. It is better to compare the proposed VC approach with SOTA VC models such as YourTTS, although YourTTS is trained with the need of text scripts.


**Summary Of The Paper:**

The paper studies the important problem of learning text-to-speech synthesis from noisy unpaired data. To order to achieve this, the authors propose several practical tricks, including normalizing variable and noisy information in speech data, curriculum learning,
length augmentation and auxiliary supervised learning. Experimental results on public datasets confirm the effectiveness of the proposed tricks.

**Summary Of The Review:**

Overall I think this is a good paper, that is likely to prove quite useful for the development of zero-shot or few-shot text-to-speech synthesis solution.

---

> ### Author Response · Authors · 2022-11-17
> **Response to Reviewer sG8y**
>
> **[About the reproducibility]** To make sure reproducibility, we add the core model implementation and training codes to the updated supplementary materials. For the full source codes, we are still doing some cleaning and simplification. We will release them after the paper acceptance.
>
> **[About comparison with other SOTA VC models]** Self-supervised representations-based voice conversion (SSL-based VC) model is currently one of the state of the art voice conversion method [1,2]. Based on the architecture of SSL-based VC, we employ multilingual HuBERT to tackle the multilingual challenge in our scenario, variational encoder and information bottleneck to improve model robustness and disentanglement ability. Compared with PPG or TTS-based voice conversion methods (e.g., YouTTS [3]), they rely heavily on paired audio-text data in the target language, which is unavailable in our settings. Compared with other unsupervised speech disentanglement methods (e.g., AUTOVC [4], SpeechSplit [5], SpeechSplit 2 [6]), we find they need very careful hyper-parameter tuning and cannot achieve better audio quality compared with the SSL-based VC model on our CommonVoice dataset which is extremely low-quality.
>
> > References:
> > [1] Polyak A, Adi Y, Copet J, et al. Speech resynthesis from discrete disentangled self-supervised representations
> > [2] van Niekerk B, Carbonneau M A, Zaïdi J, et al. A comparison of discrete and soft speech units for improved voice conversion
> > [3] Casanova E, Weber J, Shulby C D, et al. Yourtts: Towards zero-shot multi-speaker tts and zero-shot voice conversion for everyone
> > [4] Qian K, Zhang Y, Chang S, et al. Autovc: Zero-shot voice style transfer with only autoencoder loss
> > [5] Qian K, Zhang Y, Chang S, et al. Unsupervised speech decomposition via triple information bottleneck
> > [6] Chan, Chak Ho, et al. SpeechSplit2.0: Unsupervised Speech Disentanglement for Voice Conversion without Tuning Autoencoder Bottlenecks.

---

### Official Review · Reviewer_G8SV · 2022-10-24

**Confidence:** 4
**Correctness:** 4
**Technical Novelty And Significance:** 3
**Empirical Novelty And Significance:** 4
**Recommendation:** 8

**Clarity, Quality, Novelty And Reproducibility:**

The paper is very clearly written and easy to follow. Nevertheless, due to the large number of moving parts, when reading for the first time I was a bit confused about all the method applied. Not sure how to fix this. Then, some images are hard to read when printed in black and white.

The paper seems to have high quality, the methods are motivated well. The approach is sound. I am slightly unconvinced by the ablation studies. The Tab. 3 shows that many of the error bars on MOS intersect making rows 1, 5, 6, 7 identical as well as rows 2, 3, 4. What are the error bars for the CER and WER? The same is true for Tab. 4, it seems that all the error bars intersect here.

The paper is novel enough. While the paper builds upon the previous work, there are important changes and a clear end goal. From listening to the samples, I am convinced that the proposed method sounds better than the baselines.

It should be possible to reproduce the paper. But I still encourage the authors to release the source code of the paper.

# Typos

- Indonesian and French share some Latin(?) alphabets
- p.4 variable information


**Strength And Weaknesses:**

# Strengths

- Despite the complexity of the methods for TTS, the paper is able to clearly explain prior and the proposed methods.
- The motivation of the work is clear and introduced well
- Judging by the samples, the model performs very well


# Weaknesses

- I found the ablation study unconvincing: the MOS error bars intersect

**Summary Of The Paper:**

The paper improves upon the unsupervised speech synthesis. The motivation of such work is to serve to low resource languages where the supervised approach might not be feasible due to lack of annotated data.

The paper proposes and implements several modifications (bag of tricks) to the existing methods to improve the performance. The tricks are: the variational normalization to tackle the noisy information; the curriculum learning; non-autoregressive TTS.

The main contribution of the paper is that they present a high-quality method for unsupervised TTS that allows training without the lexicon.

**Summary Of The Review:**

In summary, while there are some minor issues with the experimentation and the writing, the paper is very clearly written and contains some important results. I am convinced that the paper is well suited for the conference. Accept.

---

> ### Author Response · Authors · 2022-11-17
> **Response to Reviewer G8SV**
>
> **[About the MOS error bars intersection in ablation study]**
>
> In the original version of the paper, we randomly choose 20 sentences from the test set and each audio is listened by at least 20 native testers, to narrow the error bars intersection, we re-evaluate our English models which have MOS error bars intersection with our main method in Table 3 (ablation studies) with 40 sentences and 30 native testers (it is expensive and difficult to find more Indonesian testers so we only re-evaluate English model). The results are shown in the following table. We can see that the error bars intersection can be narrowed from 0.8~0.9 to 0.5~0.7 while the mean values of MOS are stable. Although there still exists a small intersection between our method and some ablation settings, we think it is safe to draw the conclusion that these techniques are effective as the mean MOS value of our method is far from the error bars of these ablation settings.
>
> | Settings | MOS (40 sentences $\times$ 30 testers) | MOS (20 sentences $\times$ 20 testers) | CER             | WER             |
> | -------- | -------------------------------------- | -------------------------------------- | --------------- | --------------- |
> | Ours     | 3.83$\pm$0.05                          | 3.82$\pm$0.09                          | 0.145$\pm$0.006 | 0.320$\pm$0.007 |
> | w/o. CL  | 3.71$\pm$0.07                          | 3.72$\pm$0.08                          | 0.225$\pm$0.008 | 0.493$\pm$0.012 |
> | w/o. Aug | 3.74$\pm$0.06                          | 3.73$\pm$0.09                          | 0.163$\pm$0.007 | 0.358$\pm$0.008 |
> | w/o. Aux | 3.71$\pm$0.07                          | 3.70$\pm$0.09                          | 0.227$\pm$0.010 | 0.472$\pm$0.011 |
>
> **[About the CER and WER error bars]** Since CER and WER are deterministic objective metrics, we need to retrain each model with 10 different seeds and calculate the error bars of CER and WER in the above table. We can see that the error bars of these settings do not intersect with that of the main model.
>
> **[About the readability when printed in black and white]** Thanks for your advice! We modify Figure 1 and Figure 2 to make them more readable when printed in black and white in revised paper.
>
> **[About the reproducibility and code open-sourcing]** We put the core model implementation and training codes in the supplementary materials. For the full source codes, we are still doing some cleaning and simplification. We will release them after the paper acceptance.
>
> **[About typos]** Thanks for your advice! We fix these typos in revised paper.

---

### Official Review · Reviewer_oNy6 · 2022-10-24

**Confidence:** 4
**Correctness:** 3
**Technical Novelty And Significance:** 2
**Empirical Novelty And Significance:** 3
**Recommendation:** 6

**Clarity, Quality, Novelty And Reproducibility:**

I believe the clarity and quality of the presentation are good and easy to follow. The code is apparently available as well, which should make reproducibility straight forward. In terms of novelty, while this work achieves most of its results mostly by composing other methods and models, there actual composition itself to achieve the stated results is not trivial and the analysis is valuable.

nit: table 3 caption references LM which is not in the table, but not Aug which is in the table

**Strength And Weaknesses:**

Strengths:
- The final results are very good, especially for Indonesian
- There is an extensive ablation study showing the effectiveness of each component
- There are some innovations, such as the focus rate F

Weaknesses:
- The method is quite complex, combining several pre-trained models together with complex techniques, though it may be justified given the results achieved
- Many of these tricks can also be applied to unsupervised ASR (thus improving unsupervised TTS based on these methods), such as voice conversion, access to a high quality rich resource language close to the target language, etc, and these drive significant improvements in accuracy of this method, as shown in table 3. While the authors have made attempts to make use of these methods for the baselines (as described in appendix C2), it is not very clear how effective these adaptations were without any ablation studies, nor is it clear how much effort the authors put in to make sure the baselines were modified in a an optimal way.

Other:
- While using the focus rate F to judge the quality of back-translation results is interesting, is there any study on how well F maps to quality (e.g. using a paired dataset for evaluation of F)?
- In the given setup, speech and text are from same domain (CommonVoice dataset). Does this method still work if text and speech are from different domains? e,g, conversational speech + wikipedia text


**Summary Of The Paper:**

This paper presents an unsupervised TTS pipeline that is achieved by assembling several pre-trained models and various methods in a pipeline, along with a few new tricks, to achieve impressive unsupervised TTS results.

The main contributions of this paper is examining different aspects of the proposed pipeline and quantifying the contributions of each component, as well as showing that when properly integrated together, this pipeline can achieve impressive results on the setup chosen by the paper.

**Summary Of The Review:**

This paper proposes a pipeline method to perform unsupervised TTS that does not rely on pseudo-labels produced by unsupervised ASR. However it does rely on several pre-trained models to achieve good results (as seen in ablations) which are pipelined together to build the final system. The main value of this paper, in my opinion, is showing how each component in the pipeline contributes to the final quality of the system and, together with released code for reproducibility, is certainly valuable to the community.

---

> ### Author Response · Authors · 2022-11-17
> **Response to Reviewer oNy6**
>
> **[About the complexity of the method]** Yes, our method is quite complex. To make sure reproducibility, we add the core model implementation and training codes to the updated supplementary materials. For the full source codes, we are still doing some cleaning and simplification. We will release them after the paper acceptance.
>
> **[About the baselines]** Actually, we spend much time tuning the baselines based on the open-sourced (if available) or our reimplement codes. We find that if we do not apply those modifications in Section C.2 (e.g., NAR architecture and voice conversion techniques) on those baselines, we cannot achieve reasonable TTS results (WER can be larger than 0.8). The reason is that to simulate the real situation, we use the unpaired dataset which is very noisy and contains multiple speakers. Besides, we do not use any G2P tool. These restrictions greatly increase the difficulty of the task for baseline models.
>
> **[How well F maps to quality]** To verify the effectiveness of the focus rate $\mathcal{F}$ we propose in Section 3.4, we calculate $\mathcal{F}$ and CER on English test set in our model training process. We plot the curves to explore the correlation between them in Figure 6 in the revised version of the paper. From the figure, we can see that the focus rate $\mathcal{F}$ is negatively related to recognition accuracy, which means it is reasonable to use it as the indicator for filtering ASR transcriptions (higher $\mathcal{F}$ indicates lower CER). We add these analyses to Appendix D.3 in the revised paper.
>
> **[User unpaired text and speech from different domains]** We train our model using the audio data from the CommonVoice English subset which is the same as the original version of the paper and the text data from WMT16 English training set to make the domains of unpaired audio and text very different. We keep the test set the same as the original paper (LJSpeech subset). The results are shown in the following table. We find the performance drops a bit (~0.036 and ~0.06 increasing in CER and WER) due to the domain gap between the text and speech unpaired data, but we think the audio quality and intelligibility are still acceptable and promising in this unsupervised setting. We add these analyses to Appendix D.4 in the new version of the paper.
>
> | Dataset                              | MOS           | CER   | WER   |
> | ------------------------------------ | ------------- | ----- | ----- |
> | Commonvoice audio + Commonvoice text | 3.82$\pm$0.09 | 0.145 | 0.320 |
> | Commonvoice audio + WMT16 text       | 3.74$\pm$0.08 | 0.181 | 0.380 |
>
> **[Mistakes in table 3 caption]** Thanks for your advice! We fix the mistake in the caption of Table 3 in the revised paper.

---

### Decision · Program_Chairs · 2023-01-20

**Decision:**

Accept: poster

**Justification For Why Not Higher Score:**

Since the paper's contribution is to use the existing methods to achieve high-quality unsupervised TTS method, not propose some novel methods, my recommendation is "Accept (poster)", but can not be higher.

**Justification For Why Not Lower Score:**

The results and analysis are valuable. There is no good reason to reject it.


**Metareview: Summary, Strengths And Weaknesses:**

The paper's main contribution is presenting a high-quality, unsupervised TTS method.

The paper proposes and implements several modifications (bag of tricks) to the existing methods to improve the performance, including the variational normalization to tackle noisy information, curriculum learning, non-autoregressive TTS, etc. The primary value of this paper is showing how each pipeline component contributes to the system's final quality, which is valuable to the community.

The work's weakness is that it achieves most of its results by composing existing methods and models. But to achieve the stated results is not trivial, and the analysis is valuable. Another weakness is that all the results are achieved with pre-trained models. The paper does not provide the ablation without pre-training.

**Note From Pc:**

if the above contains the word "oral" or "spotlight" please see: "oral" presentation means -> notable-top-5% and "spotlight" means -> notable-top-25%. As stated in our emails, we are disassociating presentation type from AC recommendations